# Immunological Factors Associated with Adult Asthma in the Aseer Region, Southwestern Saudi Arabia

**DOI:** 10.3390/ijerph16142495

**Published:** 2019-07-12

**Authors:** Badr R. Al-Ghamdi, Emad A. Koshak, Fakhreldin M. Omer, Nabil J. Awadalla, Ahmed A. Mahfouz, Hussein M. Ageely

**Affiliations:** 1Department of Internal Medicine, College of Medicine, King Khalid University, Abha 61421, Saudi Arabia; 2Department of Internal Medicine, College of Medicine, King Abdul Aziz University, Jeddah 21589, Saudi Arabia; 3Department of Clinical Microbiology, College of Medicine, King Khalid University, Abha 61421, Saudi Arabia; 4Department of Family and Community Medicine, College of Medicine, King Khalid University, Abha 61421, Saudi Arabia; 5Department of Community Medicine, College of Medicine Mansoura University, Mansoura 35516, Egypt; 6Department of Epidemiology, High Institute of Public Health, Alexandria University, Alexandria 21511, Egypt; 7Department of Internal Medicine, College of Medicine, Jazan University, Jazan 45142, Saudi Arabia

**Keywords:** adult asthma, Saudi Arabia, total IgE, eosinophils, allergen-specific IgE

## Abstract

Background: The prevalence of asthma is on the rise in Saudi Arabia. Data regarding the immunological profile of asthma in adults in the Aseer region, in southwestern Saudi Arabia, have not been well studied. Objectives: Our aim was to study the immunological factors associated with sensitization to asthma among adults in the Aseer region. Methods: A cross-sectional study with a nested case control design in a 1:1 ratio was conducted on a sample of adults attending primary health care centers in the Aseer region. The study used a validated Arabic version of the International study of asthma and allergies in childhood (ISAAC) questionnaire. The presence of wheezing in the past 12 months was used as a proxy for bronchial asthma. Matched age and sex controls were selected. Both groups were tested for complete blood count (CBC), total and differential white blood cell (WBC) count including eosinophils, total immunoglobulin E (IgE) measurement, allergen-specific immunoglobulin E (IgE), and cytokine levels. Results: The present study included 110 cases and 157 age- and sex-matched controls. Rye wheat was found to be a significant outdoor sensitizing agent ((odds ratio) OR = 5.23, 95% CI: 1.06–25.69). Indoors, house dust mites *Dermatophagoides petronyssinus* (OR = 2.04, 95% CI: 1.04–3.99) and *Dermatophagoides farinae* (OR = 2.50, 95% CI: 1.09–5.75) were significant. Higher total IgE (OR = 1.84, 95% CI: 1.10–3.06) and eosinophil levels (OR = 2.85, 95% CI: 1.14–7.15) were significantly associated with adult bronchial asthma in Aseer. On the other hand, the role of cytokines was not significant. Conclusions: In the present study, certain environmental agents were found to be important with regards to sensitization to bronchial asthma in adults. Knowledge about these sensitization agents should be disseminated to health providers and treating physicians in order to enhance preventive environmental control measures and asthma management. Asthma-treating physicians in the region should be alerted to the use of targeted biological therapies in selected asthmatics with difficult-to-control courses.

## 1. Introduction

Bronchial asthma is a chronic inflammatory disease of the airways [1]. Based on the clinical manifestations and immunological criteria, the disease can be roughly divided into allergic (atopic) and non-allergic (non-atopic) asthma [2]. The immunological features of atopic asthma are reported to entail a distinct phenotype subscribing to the T helper cell type 2 (Th2) paradigm as well as increased blood and sputum eosinophilia [2,3,4,5,6]. 

In atopic individuals, sensitization to environmental allergenic triggering factors, including a number of indoor and outdoor allergens, may cause asthma exacerbations. The common indoor allergens may include house dust mites (HDMs), molds, pets, cockroaches, and rodents. Outdoor allergens include pollens and molds, while other potential non-allergenic triggers may include humidity, tobacco smoke, and air pollution [7,8]. Additional factors may relate to food, obesity, medicines, exercise, stress, and the microbial environment [9,10]. While the above may provide pointers for the observed clinical asthma phenotypes, limitations, nonetheless, do exist and the ultimate classification of such immunological phenotypes remains controversial [11]. 

In the Kingdom of Saudi Arabia (KSA), the average prevalence of asthma in adults is about 14.3%, and it is on the increase [12]. The disease is believed to be responsible for significant morbidity for both children and adults [13,14]. Moreover, the estimated cost of asthma diagnosis and treatment is also on the rise [15]. Earlier research conducted in KSA and neighboring countries has drawn parallels to the reported clinical and immunological phenotypes [16,17]. Some of these studies reported a potential link between elevated blood eosinophilia and severe or uncontrolled asthma. Additionally, several local studies documented the different sensitization patterns to aeroallergens in asthmatics among different regions of KSA [18,19,20,21].

However, most of the local studies on asthma in the KSA, particularly on its prevalence, have focused on children. It remains, therefore, that asthma in adults in KSA has not been given the same attention. Part of the reason for this is that the known symptoms and signs of the disease, such as episodic wheezing, breathlessness, cough, trouble sleeping, and chest tightness, may not be specific to asthma [22]. Many adults may have asthma but not know it as many do not have the traditional asthma symptoms, or they do not have all of the symptoms [23]. The presence of other conditions that can influence asthma signs and presentation may be another reason [24].

For adults, the link between environmental allergens, the risk for asthma, the Th2 phenotype, and its cytokine milieu is not clearly understood. At the basic level, what is known is that exposure to asthma sensitizing agents in early childhood can cause predisposition to early disease onset as well as asthma in adults [25,26]. Some have even questioned the difference between adult and childhood asthma [27]. 

Data regarding the immunological profile of adult asthmatics in KSA and in the Aseer region, southwestern KSA, in particular, are scarce. Hence, the aim of the present research is to study the immunological factors associated with adult bronchial asthma in the Aseer region.

## 2. Materials and Methods

### 2.1. Design

A cross-sectional study with a nested case control design in a 1:1 ratio was conducted on a sample of adults in the Aseer region, southwestern Saudi Arabia. Out of 960 surveyed adults, we identified 184 adults reported having wheeze in the past 12 months. Of the 184 adults, 110 subjects agreed to give a blood sample for further analysis. These cases were matched by age and sex in a 1:1 ratio. Individual matching was used to identify controls. Of the 776 with negative response to the International study of asthma and allergies in childhood (ISAAC) questionnaire, 157 were matched and agreed to give a blood sample for analysis.

The research proposal was approved by the Research Ethics Committee of the College of Medicine, King Khaled University, Code No. REC # 2014-06-11.

### 2.2. Description of the Study Area

The Aseer region extends from the Sarawat mountain chain down to the eastern coast of the Red Sea. To the west of these mountains is a very narrow Red Sea coastal plain, “Tehama”, which is a hot and humid area for most of the year. Health care facilities in the region include 23 hospitals and 247 primary health care centers (PHCCs).

### 2.3. Target Population

Adult males and females who attended any of the selected PHCCs for any reason were the target group of the study. The definition of an adult is “someone who is 20 years old or over”. The exclusion criteria included individuals with more than 20 pack-years of smoking and patients who were already diagnosed with chronic lung disease, bronchiectasis, chronic obstructive pulmonary disease (COPD), Interstitial Lung Disease, or prior TB infection.

### 2.4. Study Sites and Sampling

Five PHCCs were randomly selected. The selection of these PHCCs took into account urban, rural, low-, and high-altitude locations. The selected centers were Al Manhal (urban at an altitude of 2300 meters above sea level), Al Mowazafin (urban at an altitude of 2300 meters), Muhayel (urban at an altitude of 400 meters), Tharaban (rural at an altitude of 200 meters), and Al-Sooda (urban at an altitude of 3200 meters). The World Health Organization (WHO) manual for sample size determination in health studies [28] was used. The calculation was based on a conservative anticipated proportion of 20.2% of positive allergen-specific immunoglobulin E (IgE) for HDM among the general population [29] and an absolute precision of 5% at a 95% confidence interval. The minimal sample size required for the study was calculated to be 254 adults.

### 2.5. Field Visits and Participant Consent

Scheduled visits to the selected centers were arranged by the study field teams. During such visits, adult males and females attending the selected PHCCs for any reason were invited to participate in the study. A signed informed consent was obtained from each subject before inclusion in the study.

### 2.6. Data Collection

The study used a validated Arabic version of the International study of asthma and allergies in childhood (ISAAC) questionnaire that was modified to fit our adult survey. The questionnaire was previously compared to the European Community Respiratory Health Survey (ECRHS) [30,31] and found to be adequately valid. The “presence of wheeze in the past 12 months”, as stated in the ISAAC questionnaire for asthma, was used as a proxy for bronchial asthma. Individuals with negative response to the ISAAC questionnaire were considered as “not fitting the criteria of Asthma diagnosis”. Nested age- and sex-matched controls in a ratio of 1:1 were drawn from individuals with negative response to the ISAAC questionnaire. Cases and controls underwent similar laboratory investigations.

### 2.7. Blood Serum Collection and Processing

Blood was withdrawn by venipuncture from individuals included in the study. The collected blood samples were centrifuged, followed by serum isolation. The collected sera were kept frozen at −80 °C and thawed before the time of the assay. Total IgE, allergen-specific IgE, and cytokine assays were done in the immunology unit of the microbiology department at the college of medicine, KKU, Abha.

### 2.8. Complete Blood Count (CBC)

EDTA-anticoagulated blood was used for the complete blood workup. Complete blood count (CBC) analysis was done shortly after blood extraction. An automated hematologic analyzer (Beckman-Coulter), available within the existing primary health care centers, was used to determine the CBC and the total and differential white blood cell (WBC) count including eosinophils.

### 2.9. Total IgE

An ELISA kit, Human Diagnostics, Germany, catalogue number # 51015, was used for total IgE measurement. This test was used for patient screening for the presence of an allergic reaction and may help in differentiation between allergic and non-allergic asthma. The test used the World Health Organization (WHO) IgE calibration standards as a benchmark.

### 2.10. Allergen-Specific ELISA

#### Allergen Selection

For specific IgE assays, a panel of 30 common aeroallergens was selected from a review of the literature. The aeroallergens in the panel selected were mostly inhalant indoor household and outdoor allergens. The common aeroallergen panel included house dust mites, molds, and animal sources of allergens. Outdoor aeroallergens consisted of grass, trees bark, pollens, and fungus-derived spores. These allergens were selected by the investigator team based on previous research efforts to identify common allergens that are endogenous and are part of the natural fauna and flora of the study area [32,33]. Previous local studies in the KSA have shown a link between the presence of such aeroallergens and the assessment of sensitization in bronchial asthma and/or other atopic conditions [34,35,36,37,38,39,40]. 

### 2.11. Allergen-Specific IgE Immunoassay

A customized kit was used for allergen-specific IgE measurements with almost the same types of aeroallergens as previously selected. The allergen panel included in the assay kit was prepared, according to our specifications, by the manufacturing company (Mediwiss-Alleisa screen®, Moers, Germany). Measurement of serum allergen-specific IgE was done using a sensitive enzyme immunoblot assay following the manufacturer instructions (Mediwiss-Alleisa screen ®, Moers, Germany). The cut-off values for ELISA positivity were based on the kit manufacturer’s recommendation.

### 2.12. Cytokine ELISA

Serum cytokine measurements were done using sandwich ELISA kits (Invitrogen™ ELISA kits—Applied Biosystems, Houston, Texas, USA Levels of IL-4, IL-10, and IL-13 and interferon-gamma (IFN-γ) were determined according to the manufacturer’s recommendations. All assays for cytokines used pre-tittered matched pairs of coating and detection antibodies in order to achieve the accurate and sensitive measurement of individual cytokines. The reported Kit sensitivity is <1 pg/mL for IL-10, 2 pg/mL for IL-4, 0.7 pg/mL for IL-13, and 0.03 IU/mL for IFN-γ. Measurement of cytokines was done according to the protocol provided by the manufacturer for each cytokine. The cut-off values for cytokine positivity were based on the kit manufacturer’s recommendation for each cytokine.

### 2.13. Data Analysis

The compiled data were validated and analyzed using SPSS Software version 22 (IBM Corp, Armonk, NY, USA). Frequency univariate analysis was used to identify potential immunological factors associated with bronchial asthma (BA). Odds ratio (OR) and concomitant 95% confidence intervals (95% CI) were calculated.

## 3. Results

### 3.1. Description of the Study Sample

The present study included 110 cases of adults with bronchial asthma who reported wheezing in the past 12 months according to the ISAAC questionnaire and 157 age- and sex-matched controls of adults with negative response to the ISAAC questionnaire.

### 3.2. Immunological Factors Associated with Adult Bronchial Asthma

Table 1 shows the distribution of total IgE across the presence of asthma. The table shows that persons with an increase in total IgE (>100 IU/mL) had significantly higher probability (OR = 1.84, 95% CI: 1.10–3.06) to develop adult asthma. Similarly, those with an increase in total peripheral Eosinophil count (>150 cells/mm^3^) had more than two times the risk to have adult asthma (OR = 2.85, 95% CI: 1.14–7.15).

For the cytokine levels, there was an apparent higher level of IL-4 positivity among asthmatics (10.7%) compared to non-asthmatics (4.4%). Conversely, there was an inverse result recorded for IL-10 positivity, which was found to be lower in asthmatics (3.8%) compared to non-asthmatics (5%). However, these differences with regards to cytokines IL-4 and IL-10 among asthmatics and non-asthmatics were not statistically significant as judged by confidence interval values and OR (95% CI) for both IL-4 (OR = 2.61, 95% CI: 0.93–7.3) and IL-10 (OR = 0.76, 95% CI: 0.22–2.66).

Table 2 shows the distribution of increased cytokine (IL-4, IL-10, IL-13) positivity levels among asthmatics with regards to altitude and urban versus rural inhabitancy. The results show that there were no significant differences in cytokine levels by altitude or urban–rural differences.

Table 3 shows the distribution of outdoor-aeroallergen-specific IgE antibodies across the presence of asthma. The predominant outdoor aeroallergens with positive specific IgE antibodies were the following pollens: Ragweed in 24.5% of asthmatics, Bermuda grass in 20.9%, Russian thistle in 20%, Timothy grass in 20%, and Date palm in 20%. This table shows that adult persons with positive specific IgE antibodies to Rye grass had significantly more than five times (OR = 5.23, 95% CI: 1.06–25.69) the risk to have adult asthma. On the other hand, the rest of the tested outdoor-aeroallergen-specific IgE antibodies were not significantly associated with adult asthma.

Table 4 shows the distribution of indoor-aeroallergen-specific IgE across the presence of asthma. The predominant indoor aeroallergens with positive specific IgE antibodies were house dust mites *Dermatophagoides petronyssinus* in 21.1% of asthmatics and *Dermatophagoides farina* in 14.8%. This table demonstrates that adult persons with positive specific IgE antibodies to house dust mite, particularly *Dermatophagoides petronyssinus* allergen-specific IgE, had significantly more than two times (OR = 2.04, 95% CI: 1.04–3.99) the risk to have adult asthma. Similarly, adult persons with positive *Dermatophagoides farinae* allergen-specific IgE had significantly more than two times (OR = 2.50, 95% CI: 1.09–5.75) the risk to have adult asthma. Apart from house dust mites, the rest of the tested indoor-aeroallergen-specific IgE antibodies were not significantly associated with adult asthma.

## 4. Discussion

This study was conducted in a cohort of adults attending PHCCs in the Aseer region in order to identify those who might be asthmatic. The individuals recognized as asthmatic were identified by using a modified ISAAC questionnaire. Testing and analysis for total IgE, aeroallergen-specific IgE antibodies, serum cytokines IL-4, IL-13, IL-10, and IFN-γ levels, and total peripheral eosinophilic index were used as immunological biomarkers known to be associated with asthma [41].

Remarkably, in this cohort group, the total peripheral eosinophil counts and total IgE were found to be significantly elevated in asthmatic versus control individuals. This result is compatible with several earlier reports and, at present, eosinophilia and elevated IgE are regarded as important biomarkers for the identification of atopic asthma [42]. 

In this study, it was apparent that the positivity of the serum level of IL-4 was higher among asthmatics (10.7%) compared to non-asthmatics (4.4%). IL-4 is an important cytokine secreted from T helper type 2 cells; it is responsible for immunoglobulin class-switching, which leads to the production and synthesis of IgE, favoring development of type 2 inflammation and the atopic asthma phenotype [2,3,4,5].

On the other hand, the reverse was observed in the case of IL-10, where asthmatics showed lower positivity (3.8%) compared to non-asthmatics (5%). In previous reports, it was shown that IL-10, with its immunoregulatory capacity, may participate in regulation of the eosinophil count and serum IgE levels without a direct effect on asthma susceptibility [43]. Moreover, a certain IL-10 gene promoter polymorphism was reported to be linked to susceptibility to atopic asthma [44]. 

However, on statistical analysis, with regards to IL-4, IL-13, and IL-10, the difference between asthmatics and non-asthmatics did not reach significant levels. The undetectable levels of interferon gamma may be a reflection of the lack of an inflammatory process among our asthma cases and controls. Part of the reason for this apparent departure from previous reports [41] could be the small number of studied cases. However, other factors should also be taken into consideration. For example, in atopic asthma, while enhanced Th2 cytokine production and higher immunoglobulin E (IgE) may serve as valid indicators [45], this association, nonetheless, may be compromised in asthmatic patients due to several factors including undiagnosed asthma/COPD overlap [46,47]. Further stratification of asthmatics based on measured Th2 cytokines should be considered, and associations with the total and antigen-specific IgE levels should be tested. Yet, the small number of positive cases revealed in the study may preclude the usefulness of stratification. Genetic and environmental factors may also play a role in both adult and child asthma [48,49,50,51]. In adults, asthma can manifest its own phenotype [52,53] and the identification of asthma based on immunological phenotypes alone in otherwise normal adults is not always an easy proposition [27,54].

Previous studies have shown that asthma, being a heterogeneous disease, has many phenotypes; some of them can produce a systemic pattern that is different from the proposed, and widely reported, Th2 cytokine dominance [2]. Therefore, the asthma profile that shows elevated IL-4, IL-5, and IL-13, characteristic of severe asthma in children [55,56], cannot be readily ascribed to adults. This fact is highlighted by the phenotypic variability of asthma across age groups [57]. Another element factoring into this conundrum is the presence of intrinsic asthma with its altered phenotypes [58].

As noted in our present study, most of the individuals described as asthmatics tended to be mild asthmatics. The existence of Th2-low and non-Th2 phenotype response in asthma has been described and should be taken into consideration since individuals with mild to moderate adult-onset asthma can fall into this subset [58,59,60]. Individuals within this subset generally meet the broad criteria for asthma but exhibit less airway obstruction or hyper-reactivity compared to people with typical Th2-high asthma [4,60].

Our data showed Rye wheat to be an important outdoor sensitization factor for bronchial asthma in adults. The allergenic potential of Rye wheat is important and may be relevant in terms of both food and occupational asthma, e.g., in farmers and mill workers. In this respect, the issue of the local tradition of grinding wheat flour to prepare bread must be considered.

The data from this study show that the commonest sensitizing indoor aeroallergen worldwide is also the commonest in one-fifth of asthmatics in the Aseer region: The house dust mites, *Dermatophagoides petronyssinus* and the lesser *Dermatophagoides farinae* [8,9]. These sensitization agents are associated with the highest levels of specific IgE antibody production in exposed individuals who are at a significant risk for adult asthma [61,62]. In this respect, our results are not specific to this part of Saudi Arabia, as others have reported similar sensitization to the same house dust mites in other parts of the country, with the highest sensitization rate in the coastal cities [38,63,64]. It is curious that while exposure to house dust mites may be associated with high levels of specific serum IgE antibodies, it may not be associated with respiratory manifestations [29]. Nonetheless, the identification of such important triggering indoor and outdoor factors as revealed in our study can help in the control of symptoms and the overall management of asthma by encouraging and aiding the application of avoidance measures [65].

Finally, while the current study provides new insight into the immunological parameters of bronchial adult asthma in Aseer, one limitation is the relatively small number of identified asthmatics. Further large-scale study including defined cohorts of different asthma grades will be required to delineate the asthma phenotype for each cohort.

## 5. Conclusions

In the present study, certain environmental agents were found to be important with regards to sensitization to bronchial asthma in adults. Rye wheat was revealed to be a significant outdoor sensitizing agent. The house dust mites *Dermatophagoides petronyssinus* and *Dermatophagoides farinae* were found to be important indoor agents. 

Knowledge about these sensitization agents should be disseminated to treating physicians and health providers in order to enhance asthma management and preventive environment sanitation measures.

Among the immunological parameters, higher total IgE and eosinophil levels were significantly associated with adult bronchial asthma in Aseer. These findings should alert asthma-treating physicians in the region to the use of targeted biological therapies in asthmatics with difficult-to-control courses. 

The availability of biological treatments in the past decade has changed the therapeutic strategies for severe allergic asthma, and the study findings may help to improve asthma treatment in adults. 

The enhanced T helper type 2 (Th2) paradigm could not be asserted for adult bronchial asthma in the Asser area. Other phenotypes, e.g., non-/low-Th2, should be taken into consideration.

## Figures and Tables

**Table 1 ijerph-16-02495-t001:** The distribution of raised levels of cytokines, total immunoglobulin E (IgE), and eosinophil count among the study sample of adult asthmatics.

Cytokines (Cut-Off Values)	Bronchial Asthma by ISAAC (Wheezing in the Past 12 Months)		OR (95% CI)
	Non-Asthmatic *n/N* (%)	Asthmatic *n/N* (%)	
IL-4 (>13.1 pg/mL)	6/137 (4.4%)	11/103 (10.7%)	2.61 (0.93–7.3)
IL-10 (>1 pg/mL)	7/141 (5.0%)	4/105 (3.8%)	0.76 (0.22–2.66)
IL-13 (>87 pg/mL)	10/141 (7.1%)	7/105 (6.7%)	0.94 (0.34–2.54)
IFN-γ (>0.89 pg/mL)	0/142 (0%)	0/106 (0%)	-
Total IgE (>100 mmol/L)	84/156 (53.8%)	75/110 (68.2%)	1.84 (1.10–3.06) *
Eosinophil count (>150 cells/mm^3^)	38/69 (55.1%)	28/36 (77.8%)	2.85 (1.14–7.15) *

*n* = number positive; *N* = number tested; * Statistical significance (*p* ≤ 0.05); ISAAC: International study of asthma and allergies in childhood; OR: Odds ratio; CI: confidence interval; IFN-γ: interferon-gamma; IgE: immunoglobulin E.

**Table 2 ijerph-16-02495-t002:** The distribution of raised levels of cytokines among adult asthmatics with regards to altitude and urban versus rural inhabitancy.

Cytokine (Cut-Off Value)	Cytokine Positivity by Altitude		OR (95% CI)	Cytokine Positivity by Rural or Urban Inhabitancy		OR (95% CI)
	Low % (*n*)	High % (*n*)		Rural % (*n*)	Urban % (*n*)	
IL-4 (>13.1 pg/mL)	8 (11.3%)	3 (9.4%)	1.22 (0.31–4.96)	8 (14.0%)	3 (6.5%)	2.3 (0.58–9.38)
IL-13 (>87 pg/mL)	5 (6.8%)	2 (6.5%)	1.05 (0.19–5.73)	5 (8.9%)	2 (4.1%)	2.30 (0.43–12.44)
IL-10 (>1 pg/mL)	1 (1.4%)	3 (9.7%)	0.12 (0.01–1.28)	2 (3.6%)	2 (4.1%)	0.87 (0.12–6.42)
IFN-γ (>0.89 pg/mL)	0 (0.0%)	0 (0.0%)	-	0 (0.0%)	0 (0.0%)	-

OR: Odds ratio; CI: confidence interval.

**Table 3 ijerph-16-02495-t003:** Distribution of positive IgE antibodies specific to outdoor aeroallergens among the study sample of adult asthmatics.

Outdoor Aeroallergens	Number of Cases with Positive Specific IgE Antibodies		OR (95% CI)
	Non-Asthmatic *n/N* (%)	Asthmatic *n/N* (%)	
Acacia	9/156 (5.8%)	12/109 (11.0%)	2.02 (0.82–4.97)
Cypress	3/156 (1.9%)	1/110 (0.9%)	0.47 (0.05–4.56)
Juniper	5/156 (3.2%)	6/110 (5.5%)	1.74 (0.52–5.86)
Mesquite	23/156 (14.7%)	18/110 (16.4%)	1.13 (0.58–2.21)
Date Palm	29/156 (18.6%)	22/110 (20.0%)	1.09 (0.59–2.03)
Willow	9/155 (5.8%)	13/110 (11.8%)	2.17 (0.89–5.28)
Alfalfa	11/156 (7.1%)	15/110 (13.6%)	2.08 (0.91–4.72)
Chenopodium	28/156 (17.9%)	19/110 (17.3%)	0.95 (0.50–1.81)
Mugwort	6/156 (3.8%)	9/110 (8.2%)	2.23 (0.76–6.45)
Goat epithelium	28/156 (17.9%)	20/110 (18.2%)	1.02 (0.54–1.91)
Ragweed	32/156 (20.5%)	27/110 (24.5%)	1.26 (0.70–2.26)
Pigweed	18/157 (11.5%)	19/110 (17.3%)	1.61 (0.80–3.24)
Russian thistle	27/155 (17.4%)	22/110 (20.0%)	1.18 (0.63–2.21)
Bermuda grass	31/156 (19.9%)	23/110 (20.9%)	1.06 (0.58–1.95)
Timothy grass	28/155 (18.1%)	22/110 (20.0%)	1.13 (0.61–2.11)
Rye	2/156 (1.3%)	7/110 (6.4%)	**5.23 (1.06–25.69)** *
Aspergillus mix	0/156 (0%)	1/110 (0.9%)	-
Feather mix	0/156 (0%)	0/109 (0%)	-
Horse	0/156 (0%)	0/109 (0%)	-
Camel	0/156 (0%)	0/109 (0%)	-
Sheep	0/156 (0%)	0/109 (0%)	-

*n* = number positive; *N* = Number tested; * Significant (*p* ≤ 0.05); OR: Odds ratio; CI: confidence interval; IgE: immunoglobulin E.

**Table 4 ijerph-16-02495-t004:** Distribution of positive IgE antibodies specific to indoor aeroallergens among the study sample of adult asthmatics.

Indoor Aeroallergens	Number of Cases with Positive Specific IgE Antibodies		OR (95% CI)
	Non-Asthmatic *n/N* (%)	Asthmatic *n/N* (%)	
Alternaria alternate	1/156 (0.6%)	1/110 (0.9%)	1.42 (0.08–22.98)
Cladosporium	0/156 (0%)	3/110 (2.7%)	-
Penicillium mix	0/156 (0%)	0/109 (0%)	-
Yeast mix	5/156 (3.2%)	3/109 (2.8%)	0.85 (0.20–3.65)
Cat epithelium	9/156 (5.8%)	5/109 (4.6%)	0.78 (0.25–2.41)
Cockroach mix	6/156 (3.8%)	6/109 (5.5%)	1.45 (0.46–4.64)
Storage mite	3/155 (1.9%)	4/109 (3.7%)	1.93 (0.42–8.80)
*Dermatophagoides petronyssinus*	18/155 (11.6%)	23/109 (21.1%)	**2.04 (1.04–3.99)** *
*Dermatophagoides farina*	10/154 (6.5%)	16/108 (14.8%)	**2.50 (1.09–5.75)** *

*n* = number positive; *N* = Number tested; * Significant (*p* ≤ 0.05); OR: Odds ratio; CI: confidence interval.

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
