# Peer review of "Immunological Factors Associated with Adult Asthma in the Aseer Region, Southwestern Saudi Arabia"

_ijerph, 2019, doi:10.3390/ijerph16142495_

Round 1

Reviewer 1 Report

The aim of this project, according to the manuscript, is to “study the immunological factors associated with adult bronchial asthma in Aseer region in Saudi Arabia”. The authors identify a cohort of patients with presumed asthma and a control group and study “immune” certain characteristics that are different between the two groups.

Issues that need to be addressed:

1.    There is no description of the population studied. At least a table with simple demographics of the population (age, sex, etc) should be included. The age may also allow us to evaluate how likely was for patients with COPD to contaminate the selected population. Smoking history will also be needed here as a surrogate of the likelihood for COPD.

2.    Lines 113-114 “The presence of wheeze in the past 12 months, as stated in the ISAAC questionnaire for asthma, was used as proxy for bronchial asthma.” Was this the only question out of the ISAAC questionnaire used to select patients for the study? 

3.    I am not clear on the way that sample size was calculated. The authors state (lines 100-103) “Using the WHO manual for sample size determination in health studies [28], with a conservative anticipated proportion of 2.7% of allergen-specific IgE [29] and absolute precision of 2% at 95% confidence interval, the minimal sample size required for the study was calculated to be 253 adults.” What does this mean? Did they base the study on expected differences of sensitization between the two populations? I find that 2.7% of sensitization is very low for any group? Was that sensitization to a specific allergen? Which one? Especially as atopy in this population was much higher, although not clearly defined in the manuscript. It would be interesting to get an idea of atopic status for the two groups (i.e. how many of the subjects had at least one positive IgE test).

4.    How many patients were evaluated to identify the cohort they are using? They recruited 110 subjects with asthma. Out of how many evaluated? That number may be used as a rough idea of prevalence of asthma in this population. How about the rest of the group? How many did they need to screen to identify this age and sex matched control group? 

5.    How were the cut off points for ELISAs reported in Table 1 selected? The authors report these levels as “positivity”? What is the justification for this selection? In addition, the positivity for these assays for patients with “asthma” is extremely low and similar to the non-asthmatic cohort. What does this mean for the validity of the assay or the validity of the patient classification into “asthma vs. non asthma”? In general, the idea that they prove that Th2 immunity may not be important for asthma in this region is not well supported by data or by a focused and clear discussion.

6.    In the “Conclusion” section the authors state “Rye wheat was revealed to be a significant outdoor sensitizing agent”, possibly their only positive results along with sensitization to house dust mites. However, the authors do not discuss this at all in the “Discussion” section. Why do they think this is the case? Where does the exposure come from and what does it mean for the development of asthma?

7.    In the “Conclusion” section (lines 277-280) the authors state “For immunological parameters, higher total IgE and eosinophil levels are significantly associated with adult bronchial asthma in Aseer. These findings should alert asthma-treating physicians in the region to the use of targeted biological therapies in asthmatics with difficult-to-control course.” I am not clear what do they mean, especially with the second sentence in this paragraph. Firstly, even though associated, the cut off values are within the normal range. Second the authors refer to “difficult-to-control asthma” when it is not clear that this is the case for the population studied. I believe that this needs to be discussed and be careful with making generalizations about these issues.

Author Response

The authors would like to acknowledge the constructive comments of the reviewer. Please find point by point response to the raised comments. Changes are made in red in the revised file. Please find point by point response to the reviewer’s comments.

·        To guard against the likelihood of the selected population being contaminated by COPD patients, the adopted exclusion criteria were added to the methodology section as follows “The exclusion criteria included; individuals with more than 20 pack-years of smoking, patients already diagnosed with chronic lung disease, bronchiectasis, COPD, Interstitial Lung Disease and old TB infection.”  (Page: 3, line 95)

·        The presence of wheeze in the past 12 months as stated in the ISAAC questionnaire was the only criteria used as proxy for cases selection. This criterion is widely used and well accepted in epidemiological studies. (Wandalsen NF, Gonzalez C, Wandalsen GF, SolĂ© D. Evaluation of criteria for the diagnosis of asthma using an epidemiological questionnaire. Jornal Brasileiro de Pneumologia. 2009; 35:199-205).

·        Regarding sample size determination, the paragraph should read “The WHO manual for sample size determination in health studies [28] was used. The calculation was based on a conservative anticipated proportion of 20.2% of positive allergen-specific IgE for HDM among the general population [29] and an absolute precision of 5% at 95% confidence interval. The minimal sample size required for the study was calculated to be 254 adults”(Page: 3, line 105)

·        Regarding “How many patients were evaluated to identify the cohort”. The design section (2.1) should read “A cross sectional study with nested case control design of 1:1 ratio was conducted on a sample of adults in Aseer region, Southwestern Saudi Arabia. Out of 960 surveyed adults, we identified 184 adults reported having wheeze in the past 12 months. Of the 184 adults, 110 subjects agreed to give blood sample for further analysis. These cases were matched by age and sex, in 1:1 ratio. Individual matching was used to identify controls. Of the 776 with negative response to the ISAAC questionnaire, 157 were matched and agreed to give blood sample for analysis.” (Page: 2, line 80)

·        Regarding the cut off points for cytokines ELISAs reported in Table 1, it was based on manufacturer recommendation for each. A statement was added to the cytokine ELISA section in the methodology (2.12) (Page: 4, line 165).

·         On the issue of the Th2 immunity among asthmatics in the region, we stated that our data were not conclusive and not significant. However, the issue of other asthma phenotypes was discussed in two paragraphs (starting Page: 7, line 249).

·        The point regarding “Rye wheat as a significant outdoor sensitizing agent”, a paragraph was added to the discussion section. It should read “Our data show Rye wheat to be an important outdoor sensitization factor for bronchial asthma in adults. The allergenic potential of Rye wheat is important and may be relevant to both food and occupational asthma e.g., farmers and mill workers. In this respect, the issue of local tradition of grinding wheat flour to prepare bread, must be considered.” (Page: 7, line 262).

·        Regarding the point about immunotherapeutic, the sentence “These findings should alert asthma-treating physicians in the region to the use of targeted biological therapies in asthmatics with difficult-to-control course.” was removed. Authors do agree with reviewer that this statement is not directly relevant to the study and results. (Page: 7, line 291).

Reviewer 2 Report

This manuscript describes a cross-section study investigating the association between serum cytokine level, total IgE, indoor and outdoor aeroallergen specific-IgE level with asthma in adults from Aseer Region, Southwestern Saudi Arabia. The authors found that rye-wheat and dust mite are two significant sensitizing agents. Total IgE and eosinophil counts are significantly associated with asthma. Surprisingly, none of the cytokines measured (IL4, IL13, IL10 and IFN-g) were found significantly associated with bronchial asthma. Overall, the study design and data acquisition are adequate and the results add new information about these adult asthmatics in this area. This manuscript can be improved by addressing the following points:

1.     In all univariate models, age and sex should be adjusted. Are all the positive associations modified by sex and age?

2.     IFN-g doesn’t seems to be detected in all participants. Is this due to the limitation of the ELISA assay? IFN-g should be detectable based on previous studies. This should be discussed.

3.     Detection ranges for all ELISA assays should be specified in the method section.

4.     The use of using wheezing in the past year as proxy to asthma diagnosis should be discussed. Physician diagnosis of asthma should be used.

5.     As the authors discussed, there is Th2-high and low patients among asthmatics. Further stratification of asthmatics based on Th2 cytokines measured should be considered and associations with the total and antigen-specific IgE levels should be tested.

Author Response

The authors would like to acknowledge the constructive comments of the reviewer. Changes are made in red. Please find point by point response to the raised comments:

·        The design of the study used nested case control design and selected cases were matched by age and sex. Hence, there is no need to adjust for age and sex.

·        With regards to the point of IFN-g being undetectable, the essay included positive and negative controls which behaved as expected. A statement regarding this issue is added to the discussion. It should read “As far as the undetectable levels of interferon gamma, this may be a reflection of the lack of inflammatory process among our asthma cases and controls.” (Page: 7, line 238)

·        With regards to the point of ELISA cut off values, a sentence was added at the end of the Allergen-Specific IgE Immunoassay. It should read “The cut off values for ELISA positivity were based on the kit manufacturer recommendation”. (Page: 4, line 157) and (Page: 4, line 165) for cytokines.

·        The presence of wheeze in the past 12 months as stated in the ISAAC questionnaire was the only criteria used as proxy for cases selection. This criterion is widely used and well accepted in epidemiological studies. (Wandalsen NF, Gonzalez C, Wandalsen GF, SolĂ© D. Evaluation of criteria for the diagnosis of asthma using an epidemiological questionnaire. Jornal Brasileiro de Pneumologia. 2009; 35:199-205).

·        The reviewer point on Th2 levels is well taken by the authors. In consideration, the small number of positive cases revealed in the study may preclude the usefullness of stratification.

Round 2

Reviewer 1 Report

Authors replied appropriately to the reviewer's comments

Author Response

English edited version has been updated

Reviewer 2 Report

The authors have adequately addressed most of my concerns. However, in regards to the last point about stratification, I understand the sample size may be too small, however, this needs to be discussed in the manuscript.

Author Response

This paragraph was added to the manuscript to address this point. At line 248 page 7. It should read “Further stratification of asthmatics based on Th2 cytokines measured should be considered and associations with the total and antigen-specific IgE levels should be tested. Yet, the small number of positive cases revealed in the study may preclude the usefulness of stratification.”

English updated version was revised